An ELISA-based method for Galleria mellonella apolipophorin-III quantification

http://orcid.org/0000-0002-1370-4635 Ramírez-Sotelo Uriel
García-Carnero Laura C.
Martínez-Álvarez José A.
Gómez-Gaviria Manuela
http://orcid.org/0000-0001-6973-0595 Mora-Montes Héctor Manuel hmora@ugto.mx
Department of Biology, Universidad de Guanajuato , Guanajuato , Mexico
Ayayee Paul
Electronic publication date: 2024 Mar 15
Publication date: 2024
Volume: 12
Electronic Location ID: e17117
Received 2023 Oct 31; Accepted 2024 Feb 26
Copyright: © 2024 Ramírez-Sotelo et al.
Copyright year: 2024
Copyright holder: Ramírez-Sotelo et al.
License: This is an open access article distributed under the terms of the Creative Commons Attribution License, which permits unrestricted use, distribution, reproduction and adaptation in any medium and for any purpose provided that it is properly attributed. For attribution, the original author(s), title, publication source (PeerJ) and either DOI or URL of the article must be cited.
License URL: https://creativecommons.org/licenses/by/4.0/

Keywords: Opsonin, ELISA, Galleria mellonella, Candida albicans, Sporothrix, Escherichia coli, Virulence, Immune response

Funding: Consejo Nacional de Ciencia y Tecnología FC 2015-02-834 Ciencia de Frontera 2019-6380 Red Temática Glicociencia en Salud (CONACYT-México) Universidad de Guanajuato CIIC-044/2023 This work was supported by Consejo Nacional de Ciencia y Tecnología (FC 2015-02-834 and Ciencia de Frontera 2019-6380), Red Temática Glicociencia en Salud (CONACYT-México), and Universidad de Guanajuato (CIIC-044/2023). The funders had no role in study design, data collection and analysis, decision to publish, or preparation of the manuscript.

==============================
Mammalian models, such as murine, are used widely in pathophysiological studies because they have a high degree of similarity in body temperature, metabolism, and immune response with humans. However, non-vertebrate animal models have emerged as alternative models to study the host-pathogen interaction with minimal ethical concerns. Galleria mellonella is an alternative model that has proved useful in studying the interaction of the host with either bacteria or fungi, performing drug testing, and assessing the immunological response to different microorganisms. The G. mellonella immune response includes cellular and humoral components with structural and functional similarities to the immune effectors found in higher vertebrates, such as humans. An important humoral effector stimulated during infections is apolipophorin III (apoLp-III), an opsonin characterized by its lipid and carbohydrate-binding properties that participate in lipid transport, as well as immunomodulatory activity. Despite some parameters, such as the measurement of phenoloxidase activity, melanin production, hemocytes counting, and expression of antimicrobial peptides genes are already used to assess the G. mellonella immune response to pathogens with different virulence degrees, the apoLp-III quantification remains to be a parameter to assess the immune response in this invertebrate. Here, we propose an immunological tool based on an enzyme-linked immunosorbent assay that allows apoLp-III quantification in the hemolymph of larvae challenged with pathogenic agents. We tested the system with hemolymph coming from larvae infected with Escherichia coli, Candida albicans, Sporothrix schenckii, Sporothrix globosa, and Sporothrix brasiliensis. The results revealed significantly higher concentrations of apoLp-III when each microbial species was inoculated, in comparison with untouched larvae, or inoculated with phosphate-buffered saline. We also demonstrated that the apoLp-III levels correlated with the strains’ virulence, which was already reported. To our knowledge, this is one of the first attempts to quantify apoLp-III, using a quick and easy-to-use serological technique.

Introduction

The study of infective microorganisms and entities includes the analysis of the host-pathogen interaction, which helps to understand the pathophysiology of the disease and the host immune response. Even though we currently have in vitro, ex vivo, cell lines, and artificial tissues as alternatives to studying this interaction, animal models still contribute significantly to dissecting the complex relation of a pathogen with its host (Gyssens, 2019). Over the years, different model organisms have been used, as they provide valuable information on biological processes, and murine models are regarded as the gold standards for in vivo microbiological analysis (Cutuli et al., 2019; Gunatilake, 2018). The advantages of using these organisms include similarities to humans in cellular and tissue physiology, body temperature, metabolism, and immune responses (Cutuli et al., 2019; Gunatilake, 2018). However, in recent years ethical concerns, animal protection acts, and animal welfare initiatives have questioned and restricted the use of animals in scientific research, in addition to the fact that there are other drawbacks with murine models, such as the high cost required to feeding, breeding, and housing, along with the special biosafety conditions needed in animal houses for biocontainment of pathogenic agents (Barnoy et al., 2017; García-Lara, Needham & Foster, 2005; Ménard et al., 2021; Pereira et al., 2018). Because of these limitations and technical considerations, alternative model organisms with fewer ethical concerns, cheaper to maintain, and easy and safe to manipulate and breed, have been sought. Among these new alternative models, the greater wax moth Galleria mellonella has stood out as an invertebrate that enables the study of the pathogen-host interaction, and the analysis of antimicrobial drugs efficacy (Pereira et al., 2020).

The main advantage of G. mellonella in the study of the pathogen-host interaction is its immune response, which includes cellular and humoral components (Pereira et al., 2020). The G. mellonella cellular response is mediated by hemocytes, which are involved in phagocytosis, nodulation, encapsulation, clotting, and melanization upon pathogen recognition (Arteaga Blanco et al., 2017; Pereira et al., 2020). These immune cells also secrete soluble effector molecules into the hemolymph, which include antimicrobial peptides, insect metalloproteinase inhibitors (IMPI), lysozymes, phenoloxidase, and opsonins (Trevijano-Contador & Zaragoza, 2018; Vilcinskas & Wedde, 2002). These insect opsonins or complement-like proteins recognize and bind pathogen-associated molecular patterns, such as lipopolysaccharides, lipoteichoic acid, peptidoglycan, and β-1,3-glucans (Wang et al., 2006; Yoshida, Kinoshita & Ashida, 1996; Yu & Kanost, 2002; Yu & Kanost, 2003). Four classes of opsonins have been described in G. mellonella: apolipophorin-III (apoLp-III), peptidoglycan recognition proteins (PGRP), hemolin, and cationic protein 8 (Fröbius et al., 2000; Kim et al., 2010; Kordaczuk et al., 2022; Niere et al., 1999; Shaik & Sehnal, 2009; Sheehan, Tully & Kavanagh, 2020).

ApoLp-III was initially described as a protein involved in lipid transport in some insects, such as Manduca sexta, mobilizing diacylglycerols from the fatty body to the wing muscle, through the hemolymph (Blacklock & Ryan, 1994). Furthermore, this protein is a positive regulator of Plasmodium development in the mosquito Anopheles stephensi (Dhawan et al., 2017). In terms of structure, apoLp-III is characterized by having segments of amphipathic α-helices, which gives it the ability to bind neutral lipids in Orthoptera and Lepidoptera (Weers & Ryan, 2006). The G. mellonella apoLp-III has been studied in great detail due to its important role during the humoral immune response. The levels of this 18-kDa protein increase when the wax moth is challenged with pathogen agents and mediates cellular processes, such as encapsulation (Gotz et al., 1997; Halwani, Niven & Dunphy, 2000; Whitten et al., 2004). There is compelling evidence that suggests that the structural association of apoLp-III with lipids contributes to its immunomodulatory functions (Dettloff, Kaiser & Wiesner, 2001; Niere et al., 2001), but the lipid-bound apoLp-III is less efficient in binding to lipopolysaccharides (Wijeratne & Weers, 2019), and the binding of apolipoprotein to β-1,3-glucans occurs independent of its structural conformation and association with lipids (Whitten et al., 2004). It has also been reported that this opsonin shows high homology with mammalian apolipoprotein E (apoE), which participates in immunological processes such as LPS detoxification, promotes phagocytosis and the release of nitric oxide from platelets (Narayanaswami, Kiss & Weers, 2010; Tsai, Loh & Proft, 2016). Regarding its immunostimulatory functions, for example, it has been reported that apoLp-III promotes an increase in the antibacterial activity of the hemolymph and the production of superoxide by hemocytes (Niere et al., 1999; Tsai, Loh & Proft, 2016) and favors the activity of the antimicrobial peptide cecropin (Park et al., 2005; Tsai, Loh & Proft, 2016). It was also shown that apoLp-III acts coordinately with G. mellonella lysozyme, which allows the permeabilizing activity of lysozyme to increase against Gram-negative bacteria (Tsai, Loh & Proft, 2016; Zdybicka-Barabas et al., 2013).

Currently, different strategies are used to assess virulence in G. mellonella. These include mortality analysis using survival curves (Clavijo-Giraldo et al., 2016; Jemel et al., 2020), pathogen burden quantification (García-Carnero et al., 2020; Jemel et al., 2020; Lozoya-Pérez et al., 2020; Tsai, Loh & Proft, 2016), the health index score, which consists in observing the larvae irritability, melanization changes, and quantification of phenoloxidase activity (García-Carnero et al., 2020; Tsai, Loh & Proft, 2016); the gene expression analysis of some humoral effectors (e.g. gallerimicin, lysozyme and IMPI) or the identification of others such as apoLp-III, gloverin-like protein, PGRP-A, PGRP-B, and lysozyme by mass spectrometry (Altincicek et al., 2007; Freitak et al., 2007; Zhang et al., 2014). However, we currently lack quantitative immunological approaches to analyze humoral effectors of G. mellonella immunity.

Here, we develop an enzyme-linked immunosorbent assay (ELISA) method to detect apoLp-III in the hemolymph of G. mellonella larvae. In addition, we further validated this new tool by analyzing the levels of this humoral effector in larvae infected with different fungal and bacterial pathogens.

Materials and Methods

Strains and culturing conditions

The heterologous apoLp-III expression was performed in Escherichia coli BL21 Star (DE3) cells (Thermo Fisher Scientific, Waltham, MA, USA). Cells were grown in LB broth (0.5% (w/v) yeast extract, 1% (w/v) gelatin peptone, and 0.5% (w/v) NaCl) at 37 °C. When required, media was supplemented with 2% (w/v) bacteriological agar and 100 μg mL−1 ampicillin (Sigma-Aldrich, St Louis, MO, USA). To challenge G. mellonella, the fungal strains used were Candida albicans wild-type (WT) SC5314, kex2Δ null mutant, C. albicans kex2Δ/KEX2 mutant (strains CNA1 and CNA2, respectively) (Staniszewska et al., 2020), och1Δ null mutant, och1Δ + OCH1 reintegrant control (Bates et al., 2006), Sporothrix schenckii ATCC MYA-4821, Sporothrix brasiliensis ATCC MYA-4823, and Sporothrix globosa FMR-9624 (Castro et al., 2013; Madrid et al., 2009). Propagation of C. albicans strains was performed in Sabouraud broth (1% (w/v) meat peptone and casein, and 4% (w/v) glucose) at 28 °C and 120 rpm. S. schenckii, S. brasiliensis, and S. globosa yeast-like cells were obtained by inoculating conidia in YPD, pH 7.8 broth (1% (w/v) yeast extract, 2% (w/v) gelatin peptone, and 3% (w/v) dextrose), and incubating at 37 °C and 120 rpm for 4 days, as previously reported (Martínez-Álvarez et al., 2017). Infection of G. mellonella with E. coli was performed with strain DH5α (Thermo Fisher Scientific, Waltham, MA, US) grown in LB broth at 37 °C and 120 rpm. To prepare cell homogenates, yeast-like cells were suspended in 1 mL of 5 mM EDTA, 2 mM DTT, 25 mM Tris-HCl, pH 7.5, and lysis was performed with 425–600 µm size glass beads (Sigma-Aldrich, St. Louis, MO, USA), shaking 1 min in a vortex and incubating 1 min on ice, until complete 10 cycles. Then, cell homogenate was centrifuged at 9,485 × g for 10 min, and the supernatant was recovered and stored at −20 °C until use.

Recombinant expression of Galleria mellonella apoLp-III in Escherichia coli

Total RNA was isolated from G. mellonella larvae by TRIzol® reagent protocol (Invitrogen, Carlsbad, CA, USA) according to the manufacturer’s specifications. The cDNA was synthesized and purified by adsorption chromatography following the methodology reported elsewhere (Trujillo-Esquivel et al., 2016). The open reading frame from apoLp-III was amplified by PCR using primer pairs 5′-GAATTCCAAGCGGGCATAGTGCG-3′ and 5′-AAGCTTCTGCTTGCTGGCGGC-3′ (underlined bases indicate EcoRI and HindIII sites, added to the forward and reverse primer sequences, respectively) (Niere et al., 1999). The 510-bp amplicon was cloned into pJET1.2/blunt (Thermo Fisher Scientific, Waltham, MA, USA), and then subcloned into the EcoRI and HindIII sites of pCold I (Takara Bio Inc, Kusatsu, Shiga, Japan), generating pCold-apoLpIII. For recombinant gene expression, the construction was used to transform E. coli BL21 (DE3) Star competent cells (Thermo Fisher Scientific, Waltham, MA, USA) and were grown in LB broth with 100 μg mL−1 ampicillin (Sigma-Aldrich, St. Louis, MO, USA) for 20 h at 37 °C and orbital shaking (180 rpm). Then, 200 mL of fresh LB broth contained in a 2 L Erlenmeyer flask was inoculated with 1.0 mL of the overnight culture and incubated at 37 °C until reaching an OD600nm = 0.6, then isopropyl-β-D-1-thiogalactopyranoside was added to reach a final concentration of 1.0 mM and further incubated for 20 h at 15 °C. Cells were harvested by centrifuging for 20 min at 1,500 × g and 4 °C and kept at −20 °C until use.

Recombinant protein purification

Induced bacterial cells were resuspended in 5 mL of buffer A (100 mM NaH2PO4, 10 mM Tris-HCl, 8 M urea, pH 8.0), and lysis was performed with ≤106 µm size glass beads (Sigma-Aldrich, Burlington, MA, USA), shaking 1 min in a vortex and incubating 1 min on ice, until complete five cycles. Then, the sample was subjected to three cycles of freezing at −70 °C, resting at 35 °C between cycles. The cell homogenate was centrifuged at 9,485 × g for 10 min, then, the supernatant was recovered and loaded onto the Poly-Prep column (Bio-Rad, Hercules, CA, USA) with a 10 mL capacity for sample reservoir, containing 2 mL of TALON Metal Affinity Resin (Jena Bioscience, Jena, Germany). After a 60 min interaction at room temperature, the column was washed with buffer A at pH 8.0, 7.5, and 7.0 (three volumes for each pH solution), and the proteins of interest were eluted with five volumes of buffer A at pH 6.5, five volumes of the same buffer at pH 5.4 and five volumes of buffer A at pH 4.5. For each wash and elution, 2 mL fractions were collected. For purification analysis, 40 µL aliquots of each fraction were subjected to 12% (w/v) polyacrylamide gel and electrophoretically separated under denaturing conditions at 100 V for 90 min. Then, gels were washed three times with deionized water and stained with 0.25 M KCl and 1 mM dithiothreitol to visualize the protein bands and to slice out those of interest, following the procedures described previously (García-Carnero et al., 2021; Hager & Burgess, 1980). The recombinant protein was recovered through passive diffusion from polyacrylamide slices by suspending the gel pieces in PBS and incubating at 4 °C overnight with gentle shaking (120 rpm) (García-Carnero et al., 2021). For purity analysis, protein aliquots were separated by SDS-PAGE in 12% (w/v) gels and stained with silver nitrate as reported (Chevallet, Luche & Rabilloud, 2006). Protein determination was performed with a DC Protein Assay kit (Bio-Rad, Hercules, CA, USA) according to the manufacturer’s instructions.

Generation of polyclonal antibodies against recombinant apoLp-III

A New Zealand rabbit was immunized intramuscularly with an antigenic preparation that contained 150 µg of recombinant apoLp-III purified and the complete Freund’s adjuvant (Thermo Fisher Scientific), following a standard protocol (Mora-Montes et al., 2008). After 2 weeks, one booster dose (150 µg of recombinant apoLp-III purified and the incomplete Freund’s adjuvant) was injected intramuscularly, and the procedure was repeated until three booster doses were completed. A total of 2 weeks after the last booster, the rabbit was bled, the serum was collected, and the globulin fraction was precipitated with 76% (w/v) ammonium sulfate. One Aliquot of 10 mL blood was extracted from the rabbit before the immunization protocol, and the serum was collected and kept at −20 °C until used. Indirect-ELISA was used to title the antibody production, using Clear Flat-Bottom Immuno Nonsterile 96-Well plates (Thermo Fisher Scientific, Waltham, MA, US) coated with 3.0 μg recombinant apoLp-III. Briefly, wells were coated with 3 μg of pure rapoLp-III protein and incubated for 3 h at room temperature. Subsequently, the plate was washed with PBS-0.05% (v/v) Tween 20 3 times (all washes were done in this way) and blocking with 5% (w/v) casein was performed overnight at 4 °C. After washing, different dilutions of anti-apoLp-III or preimmune serum were placed from 1:100 to 1:51,200. It was then incubated for 2 h at room temperature and washing was performed. Subsequently, goat anti-rabbit IgG-HRP (Catalog number A0545; Sigma-Aldrich, St. Louis, MO, USA) was placed at a dilution of 1:2,000, incubated for 2 h at room temperature, and washed. The development was carried out with 100 μL of the substrate 3,3′,5,5′ tetramethylbenzidine (GoldBio, Olivette, MO, USA) solution was added to each well (6 mg mL−1 3,3′,5,5′ tetramethylbenzidine and 2.6 μL of hydrogen peroxide in 12 mL of a 0.1 M citrate-phosphate solution) and left incubating at room temperature in the dark for 20 min. The reaction was stopped with 50 μL of 3 M H2SO4 per well, and the absorbance at 450/540 nm was measured.The title of the antibodies generated was 1:3,200.

Total protein extraction from G. mellonella moths

Moths were obtained from an in-house colony previously established and maintained on a diet based on corn bran and honey (Clavijo-Giraldo et al., 2016) G. mellonella moths were incubated at −20 °C for 20 min. Immediately after, one moth was submerged in 500 μL PBS, pH 7.4, and vigorously homogenized with a Potter. Samples were incubated on ice for 5 min, and an additional 500 µL of PBS was added and shaken in a vortex for 30 s. The samples were centrifuged for 10 min at 8,000 × g and the supernatants were recovered. This was further centrifuged for 10 min at 15,000 × g and the supernatants were recovered in new tubes. The protein aliquots were stored at −20 °C until used.

Immunodetection of native apoLp-III by western blotting

Aliquots containing 40 µg of recombinant apoLp-III or total protein from G. mellonella moths were separated by SDS-PAGE in 12% (w/v) polyacrylamide gels at 120 V for 90 min, and then electrotransferred onto a nitrocellulose membrane Protran Premium 0.45 µm NC (Sigma-Aldrich, St. Louis, MO, USA) using a Mini Trans-Blot Cell system (Bio-Rad, Hercules, CA, US), and transfer buffer (25 mM Tris, 192 mM glycine, and 20% (v/v) methanol). Membranes were blocked with 5% (w/v) casein in PBS-Tween 20 at 0.05% (v/v) overnight at 4 °C and then incubated for 2 h at room temperature with the anti-apoLp-III antibodies at a dilution of 1:3,200, washed with PBS-Tween 20 at 0.05% (v/v) incubated for 2 h with the goat anti-rabbit IgG-HRP antibody (Catalog number A0545; Sigma-Aldrich, St. Louis, MO, USA) at a 1:2,000 dilution, and the membrane was washed again with PBS-Tween 20 at 0.05% (v/v). Color development was carried out with 1 mg mL−1 3, 3′-diaminobenzidine (Sigma-Aldrich, St. Louis, MO, USA), and 1% (v/v) hydrogen peroxide (Sigma-Aldrich, St. Louis, MO, US). The membrane was dried and the image was captured with the ChemiDoc™ MP (Bio-Rad, Hercules, CA, USA). As a control of the antibodies, an immunoblotting was performed with the pre-immune serum as the first antibody.

Infection assays in Galleria mellonella

Yeast-like cells were inoculated using cell suspensions adjusted at 2 × 109 cells mL−1 when it was used C. albicans WT, mutants and reintegrants strains; or 1 × 107 cells mL−1 when it was used S. schenckii, S. brasiliensis and S. globosa as reported (García-Carnero et al., 2020); while bacteria were inoculated from a suspension at 1 × 1011 cell mL−1 (Zdybicka-Barabas & Cytryńska, 2011). Groups containing 10 larvae were injected with 10 µL of the cell suspensions with a Hamilton syringe equipped with a 26-gauge needle in the last left pro-leg, previously sanitized with 70% (v/v) ethanol. As a control, one group was inoculated with PBS. Larvae were housed and incubated at 37 °C as reported (García-Carnero et al., 2020). After inoculation, larvae were incubated for 2, 8, 12, or 24 h, and the hemolymph was extracted (20 µL per larva), mixed in batch with 500 µL of cold PBS buffer with sodium citrate at 3.8% (w/v), giving a final dilution of the larva tissue of 1:3.5, and stored at −20 °C until used.

Quantification of Galleria mellonella apoLp-III by enzyme-linked immunosorbent assay

Clear Flat-Bottom Immuno Nonsterile 96-Well plates (Thermo Fisher Scientific,Waltham, MA, USA) were sensitized with 100 μL per well of serial dilution of the polyclonal antibodies anti-apoLp-III. The sera was diluted in 50 mM Tris-HCl, 150 mM NaCl, 1% (w/v) bovine serum albumin, pH 7.4. The plate was incubated for 2 h at 37 °C, then it was washed three times with 300 μL per well of PBS-Tween 20 at 0.05% (v/v). and then blocked with 300 μL of porcine skin gelatin in TBS at 1% (w/v) per well and incubated overnight at 37 °C. Then, plates were washed three times with PBS-Tween 20 at 0.05 % (v/v), 100 μL of the hemolymph was placed in each well and incubated at room temperature for 2 h and washed three times with PBS-Tween 20 at 0.05% (v/v). Next, 100 µL of anti-apoLp-III antibody diluted 1:400 were placed in each well and incubated for 2 h at room temperature, washed with PBS-Tween 20 at 0.05% (v/v), 50 μL of a dilution 1:2,000 of the goat anti-rabbit IgG-HRP (Catalog number A0545; Sigma-Aldrich, St. Louis, MO, USA) were placed and incubated at room temperature for 1 h. After washing with PBS-Tween 20 at 0.05% (v/v), 100 μL of the substrate 3,3′,5,5′ tetramethylbenzidine (GoldBio, Olivette, MO, USA) solution was added to each well (6 mg mL−1 3,3′,5,5′ tetramethylbenzidine and 2.6 μL of hydrogen peroxide in 12 mL of a 0.1 M citrate-phosphate solution) and left incubating at room temperature in the dark for 20 min. The reaction was stopped with 50 μL of 3 M H2SO4 per well, and the absorbance at 450/540 nm was measured. As a positive control, 100 μL of pure recombinant apoLp-III at 1.25 pg µL−1 were included instead of the hemolymph. As an extra control, the same ELISA assay was realized, but with pre-immune serum dilutions and 100 μL of pure recombinant apoLp-III at 1.25 pg µL−1 as antigen.

Statistical analysis

Infection assays in Galleria mellonella were performed in triplicate, as well as, the sandwich ELISA-like. The Dunnett’s one-way analysis of variance was used to compare the apoLp-III concentration of all larvae groups vs. the control group, or Tukey’s multiple comparison tests when comparing all pairs of groups. The latter was also used to compare ELISA readings using different antibody concentrations. All P values less than 0.05 are considered significant. The GraphPad Prism® 6.0 program was used for the statistical analysis (GraphPad Software, La Jolla, CA, USA).

Results

Heterologous expression and purification of apoLp-III in Escherichia coli

To generate an immunodetection system capable of detecting G. mellonella apoLp-III, we first generated a recombinant version of this protein in a bacterial system, as this has been previously performed and the recombinant product retained functional activity (Niere et al., 1999), suggesting that posttranslational modifications, if present in the functional polypeptide are dispensable for both lipid binding and opsonin activity (Niere et al., 1999). Since the native polypeptide contains a putative signal peptide at the N-terminus, we did not include the first 16 amino acids in the recombinant protein, and instead, the expression vector adds a TEE motif, a His tag, and a factor Xa site element. As a consequence, the recombinant version of apoLp-III is expected to be a 21 kDa polypeptide, instead of the 18-kDa native protein (Niere et al., 1999). When bacteria were transformed with pCold-apoLp-III and gene expression was induced with 1.0 mM isopropyl-β-D-1-thiogalactopyranoside for 20 h at 15 °C, a differential 21-kDa protein band was observed in cell homogenates of induced bacteria (Fig. 1A). Since this protein was absent in the cell homogenates of non-transformed cells or cells harboring the empty pCold-I vector (Fig. 1A), it was suggested that recombinant apoLp-III was generated. The recombinant protein was purified by metal affinity chromatography as detailed in ‘Material and Methods’, and the aliquots enriched with the recombinant protein were analyzed by denaturing SDS-PAGE in 12% (w/v) gels. Since both Coomassie blue and silver stainings showed only one protein band in the purified fractions (Figs. 1B and 1C), we considered the protein preparation at a purity level enough to use it for antibody generation. It is worth noting that since no 2D-electrophoretic analysis was performed, we cannot claim the protein was purified to homogeneity.

Figure 1 Expression of Galleria mellonella apolipophorin-III in Escherichia coli and its purification.

In (A) non-transformed (NT lane) E. coli BL21 cells or transformed with pCold-apoLp-III (rapoLp-III lane), or with the empty vector (EV lane) were grown under inducing conditions 1.0 mM isopropyl-β-D-1-thiogalactopyranoside and 20 h at 15 °C. Protein homogenates were prepared from induced cells and separated by SDS-PAGE in 12% (w/v) gels. Only the clone expressing pCold-apoLp-III showed a differential protein band of ~21 kDa (marked with an arrow), an expected molecular weight for the recombinant apolipophorin-III (rapoLp-III). The heterologous protein was purified as described in materials and methods and was Coomassie Brilliant Blue-stained (B) or silver-stained (C). Mw, molecular protein markers.

Production of polyclonal anti-apoLp-III antibodies

The purified protein was used as an antigen to generate rabbit anti-apoLp-III antibodies. To assess the ability of these antibodies to detect native apoLp-III in the G. mellonella hemolymph, a G. mellonella moths homogenate was prepared and used in a western blot analysis. Despite the hemolymph electrophoretic analysis showing several protein bands of different molecular weights, the antibodies only recognized one 18-kDa protein band, a molecular weight expected for native apoLp-III (Figs. 2A and 2B). As a positive control, we also detected the recombinant apoLp-III, identifying a robustly detected protein band (Fig. 2B). As negative controls, the western blot analyses were performed using the preimmune serum instead of the primary antibody, or the secondary antibody was omitted (Figs. 2C and 2D). The western blot with the preimmune serum showed a faint detection of two protein bands in the homogenate, with molecular weights of 50 and 31 kDa (Fig. 2C). A closer analysis of the results shown in Fig. 2B indicated that these are also barely detected in the homogenate by the anti-apoLp-III antibodies; however, this detection could be associated with antibodies already circulating in the rabbit sera. Nevertheless, densitometric analysis of the bands shown in Fig. 2B indicated that more than 98% of the protein detected in the hemolymph was the native apoLp-III. The western blot assay where no secondary antibody was included showed no protein band detected (Fig. 2D). Next, using ELISA as described in the Material and Methods section, the antibody sensitivity was analyzed, using different antibody dilutions and recombinant apoLp-III concentrations. When 1.25 or 0.5 µg recombinant protein was used as antigen and different dilutions of the antibodies, concentration-dependent readings were observed (Fig. 3). Except for the readings obtained with the dilution 1:100, the readings with 0.1 µg of recombinant apoLp-III were similar to those of the system with no recombinant protein, indicating our system cannot detect this amount of recombinant apoLp-III (Fig. 3). Since the reading with a dilution of 1:1,600 and 0.5 µg recombinant protein was the last that gave a statistically different absorbance this was considered the detection limit (Fig. 3).

Figure 2 Detection of native and recombinant apolipophorin-III by immunoblotting.

Preparations of 40 μg of recombinant protein (rapoLp-III lanes) and Galleria mellonella moth’s homogenates (GmH lanes) were separated by SDS-PAGE in 12% (w/v) gels, which were either stained with Coomassie Brilliant Blue (A), or were electrotransferred onto a nitrocellulose membrane, which was incubated with anti-rapoLp-III and anti-rabbit IgG-HRP (B). As an antibody control, the assay was repeated but preimmune serum was used instead of polyclonal anti-rapoLp-III (C), or the secondary antibody was omitted (D). In (B), the signal of ~18 kDa (marked with an arrow) corresponds to the molecular weight of the native apoLp-III. Mw, molecular protein marker.

Figure 3 Sensitivity of anti-rapoLp-III antibodies.

Different concentrations of recombinant apoLp-III (rapoLp-III) (given in the upper part of the figure with different symbols) were immunodetected by ELISA. The assays were performed with different dilutions of anti-rapoLp-III antibody, from 1:100 to 1:51,200), which were used to sensitize a 96-well plate. Then, rapoLp-III was added, the anti-rapoLp-III antibody was placed at the same corresponding dilution, and peroxidase-conjugated anti-rabbit IgG was included, before enzyme detection. Results are means ± SD of three independent experiments performed in duplicate. *P = 0.0002 when compared 0.5 μg mL-1 with 0.1 μg mL-1 of rapoLp-III at a 1:1,600 dilution. †P = 0.0002 when compared 0.5 μg mL-1 with 0.0 μg mL-1 of rapoLp-III at a 1:1,600 dilution. The one-way analysis of variance by Tukey’s multiple comparison test was used to establish statistical significance.

Generation of a system to immunodetect Galleria mellonella apoLp-III

The polyclonal anti-apoLp-III antibody was used to generate a Sandwich-like ELISA system to detect native apoLp-III in the G. mellonella hemolymph, as described in the Materials and Methods section. The specificity of this system was analyzed by assaying different cell homogenates of microorganisms, and we included E. coli, C. albicans, S. schenckii, S. brasiliensis, and S. globosa. In addition, we also analyzed the detection of an irrelevant recombinant protein from S. schenckii, the peptide SPSK_06559 (Giosa et al., 2020). In all cases, readings were similar to the negative control where only PBS was included, with average readings at 450 nm of 0.154 ± 0.036, which contrast with the average reading of the positive control wells containing 1.25 µg rapoLp-III (OD450 nm 1.1635 ± 0.029).

The apoLp-III response when G. mellonella larvae were infected with E. coli or C. albicans was previously analyzed by western blotting using antibodies against G. mellonella apoLp-III and a heterologous antibody against apolipophorin I and apolipophorin II from Bombyx mori (Stączek et al., 2018). They did not recognize apolipophorin III. In addition, antibodies against apoLp-III used in this study were obtained against G. mellonella apoLp-III. This analysis indicated that apoLp-III levels tended to increase in the first hour after injection of the pathogen and then decrease after 24 h of inoculation (Stączek et al., 2018). To validate our system, we used both pathogens to challenge G. mellonella larvae and then quantified the apoLp-III levels using our ELISA system. Our results indicated that the control group of larvae injected only with PBS showed a significant increment in apoLp-III levels when compared to untouched larvae (P = 0.009), referred to here as the naïve group (Fig. 4). At short post-inoculation times, such as 8 and 12 h apoLp-III levels significantly increased in the hemolymph of infected insects, an observation that applies to both groups, larvae challenged with E. coli or C. albicans (Fig. 4). As reported, at 24 h post-inoculation of either bacteria or yeast cells, apoLp-III levels reduced and were like those observed in larvae challenged only with PBS (P = 0.1076). A similar situation was observed at 48 h post-infection (P = 0.2074), suggesting that a second wave of apoLp-III increment was unlikely (Fig. 4). One possible explanation for this observations is the humoral response by this opsonin is no longer relevant at this time points because the shift to a cellular response mediated by hemocytes.

Figure 4 Detection of apoLp-III in the hemolymph of Galleria mellonella larvae challenged with Candida albicans and Escherichia coli by ELISA.

A 96-well plate was sensitized with anti-recombinant apoLp-III (rapoLp-III) at a working dilution of 1:400. Then, the hemolymph coming from a naïve larvae group or inoculated groups with phosphate-buffered saline (PBS), C. albicans at 2 × 107 yeast-like cells per inoculum, or E. coli at 1 × 109 CFU per inoculum were placed into the wells. The anti-rapoLp-III antibody was added again (using the 1:400 dilution) and finally, the detection was completed with peroxidase-conjugated anti-rabbit IgG before the development of the enzyme activity. As a positive control, 1.25 pg μL−1 of rapoLp-III was used. Results are means ± SD of three independent experiments performed by duplicate. Once inoculated, larvae were incubated for 8, 12, 24, or 48 h before hemolymph collection. One-way analysis of variance by Dunnett’s multiple comparison test was used for results analysis. **P = 0.009, when comparing the naïve and PBS-inoculated groups. *P = 0.001, when comparing the animal groups inoculated with either PBS or C. albicans at 8 and 12 h. †P = 0.001, when comparing the larvae groups inoculated with either PBS or E. coli at 8 and 12 h.

To further validate our system, we analyzed the apoLp-III levels stimulated in larva infected with C. albicans mutant strains with known virulence defects in both murine and G. mellonella models. The strains included in this validation were mutants lacking one or the two KEX2 alleles, which code for a Golgi-resident serine protease involved in the classic protein secretory pathway, and both mutant strains showed an avirulent phenotype when injected in G. mellonella larvae (Gómez-Gaviria et al., 2020). We also included the C.albicans och1∆ null mutant strain, which lacks a Golgi-resident α-1,6-mannosyltransferase involved in the synthesis of the N-linked glycan outer chain, and whose virulence is attenuated in this insect model (Bates et al., 2006; Pérez-García et al., 2016). Interestingly, both KEX2 mutants failed in inducing apoLp-III at 8 and 12 h post-infection, times associated with high lipoprotein levels in the hemolymph of larvae infected with the wild-type control strain (Fig. 5). Similar results were observed with the och1∆ null mutant strain, but the complementing mutant strain, where one copy of the disrupted gene was placed back into the null mutant background, restored the ability to induce apoLp-III at the same level as the wild-type control strain (Fig. 6). Therefore, our system to detect and quantify apoLp-III is at least useful to analyze the stimulation of this peptide by C. albicans.

Figure 5 Detection of apoLp-III in the hemolymph of Galleria mellonella larvae challenged with Candida albicans mutants kex2Δ/KEX2 and kex2Δ/kex2Δ by ELISA.

A 96-well plate was sensitized with anti-recombinant apoLp-III (rapoLp-III) at a working dilution of 1:400. Then, the hemolymph from larvae inoculated with either phosphate-buffered saline (PBS) or C. albicans was added and apoLp-III levels were quantified using anti-rapoLp-III antibodies and peroxidase-conjugated anti-rabbit IgG (the working dilution was 1:2,000). When larvae were inoculated with yeast cells, the doses used was 2 × 107 yeast per insect. The larva-fungus interaction was incubated for 8 or 12 h before hemolymph was withdrawn. Aliquots of 1.25 pg μL−1 rapoLp-III were used as a positive control. Results are means ± SD of three independent experiments performed by duplicate. *P = 0.0023, when comparing with the control group inoculated with PBS. One-way analysis of variance by Dunnett’s multiple comparison test was used for data analysis.

Figure 6 Detection of apoLp-III in the hemolymph of Galleria mellonella larvae challenged with Candida albicans null mutant och1Δ and reintegrant control by ELISA.

A 96-well plate was sensitized with anti-recombinant apoLp-III (rapoLp-III) at a working dilution of 1:400. Then, the hemolymph from larvae inoculated with either phosphate-buffered saline (PBS) or C. albicans was added and apoLp-III levels were quantified using anti-rapoLp-III antibodies and peroxidase-conjugated anti-rabbit IgG (the working dilution was 1:2000). When larvae were inoculated with yeast cells, the doses used was 2 × 107 yeast per insect. The larva-fungus interaction was incubated for 8 or 24 h before hemolymph was withdrawn. Aliquots of 1.25 pg μL−1 rapoLp-III were used as a positive control. Results are means ± SD of three independent experiments performed by duplicate. *P < 0.0001 when compared with the groups inoculated with PBS. One-way analysis of variance by Dunnett’s multiple comparison test was used for data analysis.

We also explored the usefulness of the apoLp-III detection system when G. mellonella larvae were inoculated with Sporothrix yeast-like cells, as these larvae have been validated to study the virulence of species of the Sporothrix pathogenic clade (Clavijo-Giraldo et al., 2016; Lozoya-Pérez et al., 2020). When S. brasiliensis, S. schenckii, or S. globosa yeast-like cells were inoculated in larvae, the three fungal species stimulated the production of apoLp-III (Fig. 7). Alternatively, the increment of apoLp-III could be due to releasing of this protein stored in fat body cells. In the groups infected with S. brasiliensis and S. schenckii the apoLp-III levels incremented after 2 h of inoculation and this was further induced at 4 h post-inoculation (Fig. 7). However, when compared to the apoLp-III levels at 4 and 8 h post-inoculation this increment was not significant (P = 0.181 and 0.417 for S. brasiliensis and S. schenckii, respectively). For the case of S. globosa, the apoLp-III quantification at 2h post-inoculation was significantly higher when compared to those levels stimulated with PBS (P = 0.0457) but this did not further increment at 4 or 8 h post-inoculation (Fig. 7). When the apoLp-III stimulation by the three fungal species was compared, the data indicated that S. brasiliensis stimulated the highest levels of this protein at 2 and 4 h post-inoculation, while S. globosa was the lowest (Fig. 7). At 8 h post-inoculation, the only statistically significant difference was observed between larvae stimulated with S. brasiliensis and S. globosa (P = 0.0143). Collectively, these data indicate that our apoLp-III detection system allowed us to generate species-specific stimulation patterns for the Sporothrix species under analysis.

Figure 7 Detection of apoLp-III in the hemolymph of Galleria mellonella larvae challenged with Sporothrix brasiliensis, Sporothrix schenckii, or Sporothrix globosa by ELISA.

A 96-well plate was sensitized with anti-recombinant apoLp-III (rapoLp-III) at a working dilution of 1:400. Then, the hemolymph from larvae inoculated with either phosphate-buffered saline (PBS) or yeast-like cells was added and apoLp-III levels were quantified using anti-rapoLp-III antibodies and peroxidase-conjugated anti-rabbit IgG (the working dilution was 1:2,000). When larvae were inoculated with yeast cells, the doses used was 2 × 107 yeast-like cells per insect. The larva-fungus interaction was incubated for 8 or 24 h before hemolymph was withdrawn. Aliquots of 1.25 pg μL−1 rapoLp-III were used as a positive control. Results are means ± SD of three independent experiments performed by duplicate. *P = 0.0039 when compared with the other animal groups at 2 h post-inoculation. **P = 0.0457 when comparing the animal group inoculated with PBS with the groups inoculated with S. globosa yeast-like cells. †P = < 0.0001 when compared with the groups inoculated with either S. schenckii or S. globosa. ††P = 0.0143 when compared with the animal group inoculated with S. globosa at 8 h post-inoculation. ‡P = 0.0039 when compared with the group inoculated with S. globosa at 4 h post-inoculation. The One-way analysis of variance by Tukey’s multiple comparison tests was used to establish statistical significance.

Discussion

G. mellonella is an attractive model for studying the pathogen-host interaction, to assess virulence, toxicity, efficacy of antifungal drugs, and immunological priming (Clavijo-Giraldo et al., 2016; García-Carnero et al., 2021; Jemel et al., 2020; Martínez-Álvarez et al., 2019; Staniszewska et al., 2016). Despite there are many indicators to determine pathogen virulence in this organism (Jemel et al., 2020), there are immunological parameters that can be included in the analysis of the G. mellonella-pathogen interaction, such as apoLp-III.

The detection system reported here was based on the production of rapoLp-III in a prokaryotic system because it was previously reported that eukaryotic post-translational modifications are not required to generate a functional protein in bacteria (Niere et al., 1999), and this protein was used to generate polyclonal antibodies whose titer was better to those previously generated (Zdybicka-Barabas & Cytryńska, 2011; Zdybicka-Barabas et al., 2015, 2012). The detection of native apoLp-III in both adults and larvae with the polyclonal antibody made it feasible to conceive its use in a quantitative immunodetection system. As mentioned, several ways have been proposed to determine the role of this protein during the G. mellonella-pathogen interaction, such as semiquantitative western blot, quantitative expression assays by RT-qPCR, or through mass spectrometry (Altincicek et al., 2007; Freitak et al., 2007; Zhang et al., 2014). However, there could be some drawbacks to these methods. For example, for RT-qPCR, it is known that the type of sample and the collection process are relevant parameters to control, as they may affect the final quantification of gene expression (Mahanama & Wilson-Davies, 2021). Moreover, the detection of gene expression alone is not conclusive for pathogenesis and usually has to be complemented with the quantification of additional parameters (Mahanama & Wilson-Davies, 2021). Regarding mass spectrometry, the main limitation is the high cost required by the technique (Chen et al., 2021).

We demonstrated that the ELISA system generated here was capable of showing different apoLp-III levels in the hemolymph of larvae infected with C. albicans or E. coli. These results suggested that the detection system can be used to measure apoLp-III concentration a few hours after infection, obtaining a fast and reliable analysis of the virulence of the tested pathogens. In addition, our results were similar to those previously reported (Stączek et al., 2018), which showed that E. coli stimulated more apoLp-III than C. albicans. To know whether the apoLp-III detection by ELISA is capable of discriminating virulent from non-virulent C. albicans strains we included in our analysis the mutants kex2Δ/KEX2, kex2Δ/kex2Δ, and och1Δ. As expected, these three mutants failed to stimulate strong apoLp-III levels, suggesting that this could be considered an early predictor of virulence, at least for this fungal species. It is worth noting that the och1∆ reintegrant strain showed the ability to kill larvae at the same ratio as the wild-type control strain (Bates et al., 2006; Pérez-García et al., 2016) and here stimulated apoLp-III levels similar to the wild-type strain, confirming the restoration of virulence. Previous studies in G. mellonella have shown that S. brasiliensis is the most virulent species, followed by S. schenckii and S. globosa (Lozoya-Pérez et al., 2020). Here, we observed a similar trend in apoLp-III levels, being the lowest associated with larvae infected with S. globosa and the highest with those inoculated with S. brasiliensis. Thus, these results once again point out that the levels of this opsonin in hemolymph could be used as virulence predictors.

The main precaution to take into account in this assay is the dilution of the polyclonal anti-apoLp-III antibodies. For this, it is recommended antibody titration by indirect ELISA when absorbance values fluctuate. Knowing the optimal working concentration of antibodies is key to success in the ELISA technique (Kohl & Ascoli, 2017). In the same line, an appropriate antigen concentration used as a control is recommended to accurately calculate the apoLp-III levels in hemolymph samples.

Conclusions

The G. mellonella apoLp-III is an opsonin that participates in the humoral immune response, with the ability to maintain or increase its hemolymph levels depending on the insect’s health status. Here, an ELISA system to detect this humoral effector was generated, and results showed that the apoLp-III levels varied depending on the pathogen and the virulence attributes. We propose the apoLp-III quantification as a new parameter to assess fungal virulence in G. mellonella.

Supplemental Information

Supplemental Information 1 Original Gels to generate Figures 1 and 2.

Supplemental Information 2 Raw data for Candida albicans och1.

Supplemental Information 3 Raw data for Candida albicans Kex2.

Supplemental Information 4 Raw data for Candida albicans.

Supplemental Information 5 Raw data of Sporothrix.

Supplemental Information 6 Raw Data for E. coli.

Supplemental Information 7 Raw data of absorbance values for specificity.

Supplemental Information 8 Raw data of absorbance values for sensitivity.

Supplemental Information 9 Absorbance values with preimmune serum.

We thank Luz A. López-Ramírez (Universidad de Guanajuato) for technical assistance.

Additional Information and Declarations

Competing Interests

Author Contributions

Data Availability

Hector M. Mora-Mones is an Academic Editor for Peer Journal.

Uriel Ramírez-Sotelo conceived and designed the experiments, performed the experiments, analyzed the data, prepared figures and/or tables, authored or reviewed drafts of the article, and approved the final draft.

Laura C. García-Carnero performed the experiments, analyzed the data, prepared figures and/or tables, and approved the final draft.

José A. Martínez-Álvarez performed the experiments, analyzed the data, prepared figures and/or tables, and approved the final draft.

Manuela Gómez-Gaviria performed the experiments, analyzed the data, prepared figures and/or tables, and approved the final draft.

Héctor Manuel Mora-Montes conceived and designed the experiments, analyzed the data, prepared figures and/or tables, authored or reviewed drafts of the article, and approved the final draft.

The following information was supplied regarding data availability:

The original images of SDS-PAGE gels and blots, and raw data are available in the Supplemental Files.

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
