# Peer review of "An ELISA-based method for Galleria mellonella apolipophorin-III quantification"

_PeerJ, doi:10.7717/peerj.17117_

## Round 0.1 · original submission · Major Revisions

After going through all three reviews, there are some concerns raised by one that adds weight to counterbalance the other two. I am inclined to render a major revisions decision on your submission. Please address these concerns and add more contextualized frameworks to the introduction.

**Language Note:** The review process has identified that the English language must be improved. PeerJ can provide language editing services - please contact us at copyediting@peerj.com for pricing (be sure to provide your manuscript number and title). Alternatively, you should make your own arrangements to improve the language quality and provide details in your response letter. – PeerJ Staff

Reviewer 1 ·

Basic reporting

1. The manuscript doesn’t include sufficient introduction and background, especially in a field of involvement of apoLp-III in immune response as free and lipophorin-bound molecule. Some prior literature items have not been properly cited, e.g. Cutuli et al., 2019; Barnoy et al., 2017; Garcia-Lara et al., 2005; Menard et al., 2021; Pereira et al., 2018 (all articles concern G. mellonella or other invertebrate models, not murine models used in in vivo studies); Wang et al., 2006; Yoshida et al., 1996; Frӧbius et al., 2000 (these articles concern β-GRPs/GNBPs and serine proteinase inhibitors which are not opsonins; also not every peptidoglycan recognition protein can be classified as an opsonin); Gӧtz et al., 1997; Halwani et al. 2000 (these articles report on immune induction after injection of purified apoLp-III, phagocytosis, and interaction of apoLp-III with LTAs); Freitak et al., 2007; Zhang et al., 2004 (these articles report on results of research conducted on Trichoplusia ni and Manduca sexta, respectively, not on G. mellonella).
There is also no proper citation for “cationic protein 8” (line 72) – please cite Kim GH et al., JBC, 2010 and Kordaczuk et al., Sci Rep, 2022.
2. The figures are relevant to the content. However, the descriptions of Figures 4-7 contain information that is inconsistent with that provided in Materials and Methods: the CFU numbers of bacteria and fungi used for immune-challenge of G. mellonella larvae and amount of rapoLp-III used as a positive control in ELISA tests. In addition, Figure 2 shows results of immunodetection of apoLp-III in homogenates obtained from G. mellonella imagines, whereas in the Results section the authors described apoLp-III detection in larval hemolymph (lines 265, 275, 278). Moreover, in Figure 2B, C, D the results of immunoblotting were presented as separate strips cut from the membrane - why aren't the entire membranes shown?
3. Although appropriate raw numerical data have been made available, photographs of multi-well plates presenting results of colorimetric ELISA tests should be provided. In addition, the results of ELISA tests conducted using the obtained anti-rapoLp-III antibodies should be confirmed using immunoblotting after SDS-PAGE electrophoresis, in which analogous samples of the hemolymph of G. mellonella larvae were separated as used in the ELISA test. The membranes after such immunoblotting should be also presented. (please see also comments on Experimental design and Validity of the findings)

Experimental design

1. I suggest to improve the description at lines 74-86 to provide more justification of the knowledge gap being filled by the results.
2. There are some important questions about the Methods to be answered:
Lines 158-159 – The purity of the recombinant apoLp-III was examined by one dimensional SDS-PAGE. Although His-tag was used for purification, there is not 100% certainty about the purity of the preparation. Please explain, why you used this method instead of 2D IEF/SDS-PAGE?
Lines 164-173 – Please provide an information about a consent of the appropriate ethics committee regarding use of animals for obtaining antibodies.
In this aspect my concern is also, that if the purified rapoLp-III was contaminated by some other proteins, the obtained antibodies may additionally contain antibodies that recognize other antigens present in rapoLp-III preparation (and of course those that are in preimmune serum).
Please provide description of preparing multi-well plates coated with rapoLp-III (including type of plates) used for determining the titer of antibodies.
Lines 176-184 – Total protein extract of G. mellonella adults was prepared and used for testing anti-rapoLp-III antibodies. How many moths were used for this puropse? Taking into account the fact that hemolymph of G. mellonella larvae contains a lot of apoLp-III and that the ELISA assay described in the manuscript was developed for determining apoLp-III content in larval hemolymph, please explain why protein extract from imagines was used instead of hemolymph from G. mellonella larvae?
Lines 186-199 – Please explain why so much rapoL-III (40 µg) was used for immunoblotting? Please provide catalog number of the goat anti-rabbit IgG-HRP antibodies used for Western blotting and for ELISA assay. The 1:2000 dilution for secondary antibodies seems to be much too low (usually 1:10,000 to 1:30,000 dilution is used). Too low dilution of secondary antibodies could lead to incorrect and misleading results.
Lines 201-210 – Please provide information about a volume of hemolymph collected from a single larva. Were the hemolymph samples pooled? What volume of hemolymph was mixed with 500 µL of PBS buffer?
Lines 213-230 – I have very serious doubts about the validity of the approach to developing this Sandwich ELISA assay using polyclonal anti-rapoLp-III antibodies. Firstly, the same antibodies were used as "capture antibodies” and "detection antibodies". For the Sandwich ELISA to work properly, these antibodies should recognize different epitopes, which is not possible in the case of the antibodies used by the authors. Secondly, antibodies against rapoLp-III were not purified from obtained preparation, so they also contain antibodies present in the preimmune serum. And all these antibodies have the possibility of interacting with secondary antibodies. Thirdly, apoLp-III can occur in the hemolymph as a free molecule, bound to lipids, and as a component of lipophorin particles. Each of these forms behaves differently and it is difficult to assess the extent to which each of them is able to be bound by the "capture antibody". Therefore, I think that the method of quantitative detection of native apoLp-III in hemolymph proposed by the authors does not reflect the actual situation. Hence my previous suggestion carrying out analogous analyzes using the Western blotting method.

Validity of the findings

The hemolymph of non-immunized and buffer- or water-immunized G. mellonella larvae contains relatively high level of apoLp-III which can be easily visualized by Coomassie Brilliant Blue staining after SDS-PAGE. Therefore, the very low amount of apoLp-III detected in the hemolymph of PBS-immunized larvae using the developed Sandwich ELISA is very puzzling. Considering that the same antibody preparation was used as "capture" and "detection" antibodies, this may be a result of insufficient binding of detection antibodies to apoLp-III due to the recognition of the same epitopes by “capture” and “detection” antibodies. Taking into account this and other my remarks, I have very serious doubts about the validity of the findings.

Reviewer 2 ·

Basic reporting

The manuscript is well-written, with clear and professional English, and the structure conforms to the Journal standards. A minor observation in line 141 where “performed” should be used instead of “performant”.
The authors provide enough and concise background, citing references that are relevant to the work, and present the results using relevant figures. A minor observation, in figure 1, the description mentions a differential protein band of ~23 kDa, but in the manuscript it is mentioned to be of ~21 kDa, please unify to avoid confusion.
As a suggestion, simplify the significance in the figure legends to P< 0.05, P<0.01, P<0.001.
All the raw data was supplied.

Experimental design

In the Materials and Methods section, regarding the Infection assays in Galleria mellonella, it is not clear if the collected hemolymph from larvae was stored individually or pooled for ELISA essays. If pooled, did the sample always contain the hemolymph of 10 larvae? Were there any deaths? This is intriguing since the cell concentration inoculated of yeast-like cells (2 x 10^5) differs from that used in the cited reference (1 x 10^5) (García-Carnero et al. 2020), and in such paper, there were deaths on day 1 using half the concentration of the inoculum reported in this work. Please elaborate.

Which concentrations were used when Candida cells were inoculated? Aggregation has been reported in mutants of C. albicans like those used in this work, how was this avoided to achieve the cell concentration and ensure proper inoculation? Please elaborate.

Validity of the findings

No comment.

·

Basic reporting

The manuscript is good, however English need to be corrected in some places for better understanding.
Title: The title need to be change. Generation is not the appropriate word and title does not reflect the theme of the work.

Abstract: line 36 "or animals inoculated with the vehicle". modify this sentence

Introduction: Line 43 Reframe the whole sentence according to the reference (Gyssens 2019)
Line 58- 59: Reframe it to remove grammatical error.
Line 100-101: "Animal" covers a lot, better to use particular insect word used in this study

Experimental design

The Experimental design is good and clearly explain recombination protein of ApoIII and infection study. Few comments are given here
Material & Methods: Line 178 again Animal word is used, change it wherever it is used as whole study was done on G. mellonella/ larvae
Line 202-203 what particular yeast was used in this experiment?
Line 210: Please write the amount of hemolymph collected and method of extraction.

Validity of the findings

Results are clear Few comments are here
Line 254 Why after adding it was kept in 20 h at 15 °C? What is the reason of using lower temperature instead of Room Temperature.
Figure 1: Figure legend need to be short and clear.
Line 266: Is it whole moth hemolymph or larvae hemolymph?
Figure 2 legend create confusion that the whole insect was homogenate or hemolymph was taken in this experiment?
Line: 274-275 ApoIII is present in both lipid free and lipid bound form, how you detect single band when whole moth extract was used in the study?
Figure 3: Check Legend it should be 1:100 dilution
Line 314-315: Though it is experimental manuscript, do you have any possible explanation of lower the ApoIII expression at 24 and 48 hour?
Figure 5: Animal word is very unappropriated in the legend and also create confusions. not clear it is used for larvae, whole insect or hemolymph?
Figure 6 Rewrite the legend

Additional comments

Here ELISA is used which showed better option then other know methods used till date. Therefore it is important to write precautions and troubleshooting while using this technique.

Is there any difference in the ApoIII antibody generated from Bombyx mori than apoLp-III of G. mellonella? Line 301-303.

---

## Round 0.2 · Minor Revisions

The reviewers have found significant improvement to the revised submission .However, there are some minor issues that they have pointed out that once clarified might make the submission suitable for acceptance. Please address these issues thoroughly and resubmit for consideration.

Reviewer 1 ·

Basic reporting

The Introduction section has been improved considerably. Now, the manuscript includes sufficient introduction and background. However, there is still no proper citation for “cationic protein 8” (line 72) – Kim GH et al., JBC, 2010 and Kordaczuk et al., Sci Rep, 2022 should be cited here, as they discovered, identified and characterized this protein.
The figures are relevant to the content. The descriptions of Figures have been corrected.
Appropriate raw numerical data have been made available.

Experimental design

The authors supplemented and corrected descriptions of the methods used. Many issues have been explained in detail in their response to reviewer’s comments. I just have a few suggestions:
Lines 113-129 – Please supplement this subsection with description of preparation of cell homogenates of different microorganisms used (mentioned in Results, lines 319-325).
Line 145 – Please indicate that the used IPTG concentration was a final concentration.
Lines 238 and 245 – Please, provide the dilution of the polyclonal antibodies – was it 1:400 as indicated in description of Figures 4-7? It is not clear. Presumably, the authors wanted to include in the description both the procedure for determining the sensitivity of the obtained polyclonal antibodies (Fig. 3) and the procedure for detecting apoLp-III in the tested hemolymph samples (Figs 4-7).
Line 243 – Please provide information that this volume of hemolymph refers to hemolymph diluted 3.5 times with PBS.
Lines 237-256 – After the authors' explanations, I still have some doubts about the proposed method. If the same polyclonal antibodies are used as "capture" and "detection" antibodies, a part of them should be labeled. It can be direct labeling and then there is no need to use tertiary antibodies. And if tertiary antibodies are used, a part of polyclonal antibodies may be labeled with e.g. biotin, while tertiary antibodies are marked with streptavidin. This causes tertiary antibodies to be able to distinguish between "detection" and "capture" antibodies and bind only to "detection" antibodies. Nevertheless, in the light of the raw data contained in the Supplementary material and the authors' explanations, I think that the adopted procedure can be considered acceptable.

Validity of the findings

In the light of the authors' explanations, all the raw data contained now in the Supplementary material, and substantial improvement made to the manuscript, I have no major doubts about the validity of the results presented.

Additional comments

1. Please use the full Latin name of the species when it first appears in the text. Then please use the abbreviated species name (e.g. Galleria mellonella and G. mellonella).
2. Line 86 – Please rewrite this sentence: “…structural conformation and association with lipids”.
3. Line 87 – Please rewrite: “…with mammalian apolipoprotein E…” or “…with apolipoprotein E (apoE) from mammals…”.
4. Line 208 – “Sample” in a title is not needed.
5. Line 278 – Which is the right molecular weight: 21 kDa or 23 kDa (in description to Fig. 1)?
6. Line 281 – Please correct into “apoLp-III”.
7. Lines 299 and 304 – Please use “homogenate” instead of “hemolymph”.
8. Line 310 – Please rewrite: “…0.1 µg of recombinant apoLp-III…”.
9. Line 322 – Please provide literature citation for the peptide SPSK_06559.
10. Lines 327-328 – Please correct this sentence. The heterologous antibodies used by Stączek et al. were against apolipophorin I and apolipophorin II from Bombyx mori. They did not recognize apolipophorin III. Antibodies against apoLp-III used in this study were obtained against G. mellonella apoLp-III.
11. Line 363 – The changes in apoLp-III level detected in hemolymph can also be caused by releasing of this protein stored in e.g in fat body cells.
12. Figure 4 – In 10th line of description, please use “collection” instead of “recollection”.
13. Figures 5-7 – In the descriptions, please use “larvae” or “larva” instead of “animal”. Also, please rewrite: “The larvae were incubated for 8 or 12 hours before…”.

Reviewer 2 ·

Basic reporting

Minor comments:

Line 58: “greater max moth” should be “greater wax moth”

In figure legend 1, the description still mentions a differential protein band of ~23 kDa, but in the manuscript it is mentioned to be of ~21 kDa, please unify to avoid confusion. Also there is an extra parenthesis after “pCold-apoLp-III”.

In figure legend 4, there is missing the decimal point at † P = 0001

In line 36, and figure legends 4, 5, 6, 7 the word “animal/s” should be amended to larvae to avoid confusion.

Experimental design

No comments

Validity of the findings

No comments

---

## Round 0.3 · Minor Revisions

You have successfully addressed various issues raised by the reviewers; please address these minor comments and issues raised now for final determination. Thank you for your input.

Reviewer 1 ·

Basic reporting

The descriptions of Figures still contain mistakes that should be corrected.
1. Figure 1 – There is still discrepancy between molecular weight in the text (line 284 – 21 kDa) and a description to the Figure - 23 kDa.
2. Figure 4 – In 10th line of description, please use “collection” instead of “recollection”.
3. Figures 5-7 – In the descriptions, please use “larvae” or “larva” instead of “animal”. Also, please rewrite: “The larvae were incubated for 8 or 12 hours before…”.

Experimental design

No comments.

Validity of the findings

No comments.

Additional comments

Lines 333-336 – Please correct. My suggestion is as follows: “The apoLp-III response when G. mellonella larvae were infected with E. coli or C. albicans was previously analyzed by western blotting using antibodies against G. mellonella apoLp-III and a heterologous antibody against apolipophorin I and apolipophorin II from Bombyx mori (Stączek et al. 2018).”

·

Basic reporting

The revised Manuscript is clear and cited all references. However the Dhawan et al 2017 may be added for the role of Apolipophorin III as positive regulator of Plasmodium Development in Anopheles stephensi

Experimental design

The Revised manuscript added all the relevant information.

Validity of the findings

The revised manuscript validated the data and made all the corrections

Additional comments

It is in better form and can be published.

---

## Round 0.4 · accepted · Accept

Congratulations on the acceptance of your submission for publication with PeerJ.